# Genetic manipulation of cell line derived reticulocytes enables dissection of host malaria invasion requirements

Timothy J. Satchwell [1,2,3,5], Katherine E. Wright [4,5], Katy L. Haydn-Smith[1,2,3], Fernando Sánchez-Román Terán[4], Pedro L. Moura [1], Joseph Hawksworth[1], Jan Frayne[1,2], Ashley M. Toye [1,2,3] & Jake Baum [4]

Investigating the role that host erythrocyte proteins play in malaria infection is hampered by the genetic intractability of this anucleate cell. Here we report that reticulocytes derived through in vitro differentiation of an enucleation-competent immortalized erythroblast cell line (BEL-A) support both successful invasion and intracellular development of the malaria parasite *Plasmodium falciparum*. Using CRISPR-mediated gene knockout and subsequent complementation, we validate an essential role for the erythrocyte receptor basigin in *P. falciparum* invasion and demonstrate rescue of invasive susceptibility by receptor re-expression. Successful invasion of reticulocytes complemented with a truncated mutant excludes a functional role for the basigin cytoplasmic domain during invasion. Contrastingly, knockout of cyclophilin B, reported to participate in invasion and interact with basigin, did not impact invasive susceptibility of reticulocytes. These data establish the use of reticulocytes derived from immortalized erythroblasts as a powerful model system to explore hypotheses regarding host receptor requirements for *P. falciparum* invasion.

[1] School of Biochemistry, University of Bristol, Bristol, UK. [2] NIHR Blood and Transplant Research Unit, University of Bristol, Bristol, UK. [3] Bristol Institute for Transfusion Sciences, National Health Service Blood and Transplant (NHSBT), Bristol, UK. [4] Department of Life Sciences, Imperial College London, London SW7 2AZ, United Kingdom. [5] These authors contributed equally: Timothy J. Satchwell, Katherine E. Wright. Correspondence and requests for materials should be addressed to T.J.S. (email: t.satchwell@bristol.ac.uk) or to K.E.W. (email: k.wright@imperial.ac.uk) or to J.B. (email: jake.baum@imperial.ac.uk)

Malaria, an infectious disease caused by *Plasmodium* parasites, is an enormous economic and health burden. Every year > 200 million clinical cases and almost half a million deaths are reported, with most fatalities occurring in children under the age of five[1]. Parasite invasion into and development within red blood cells (RBCs) is responsible for all pathology associated with this disease. Invasion begins with the interaction between a merozoite (the invasive parasite form) and the RBC surface, which precedes penetration and intracellular vacuole formation via mechanisms that remain incompletely understood. One host protein implicated in the invasion process is basigin (BSG, CD147), a surface receptor believed to be essential for invasion via its interaction with *Plasmodium falciparum* Rh5[2], though our understanding of the function the interaction plays in invasion is limited.

One of the biggest obstacles to the investigation of host protein involvement in red blood cell invasion is the intractability of this anucleate cell as a system for genetic manipulation. Elegant use of proteases, blocking antibodies, and the identification and study of rare naturally occurring red blood cell phenotypes have provided valuable information regarding the requirement for individual receptors (reviewed in ref. [3–5]). However, reliance upon the identification of often vanishingly rare blood donors to provide insight is inefficient and precludes hypothesis-driven investigation of host protein involvement in invasion.

The capacity to derive reticulocytes (young red blood cells) that are susceptible to invasion by malaria parasites through in vitro culture and differentiation of hematopoietic stem cells (CD34$^+$ cells) isolated from peripheral blood or bone marrow has opened up myriad new possibilities to erythrocyte biologists. Such cells are phenotypically equivalent to in vivo-derived reticulocytes and display functional equivalence to red blood cells[6–8]. Through lentiviral transduction of immature nucleated erythroblast precursors prior to differentiation it is now possible to generate enucleated reticulocytes with rare or novel phenotypes to study host cell protein requirements and involvement in invasion. The power of this approach was demonstrated in 2015 in a forward genetic screen employing shRNA-mediated knockdown of blood group proteins in primary in vitro-derived reticulocytes. This study identified important roles for CD55 and CD44 in *P. falciparum* invasion[9]. Although informative, shRNA-mediated depletion of gene expression frequently results in incomplete knockdowns that can mask all but the most obvious of invasion defects. Furthermore, the finite period in which transduced nucleated cells can be maintained in their undifferentiated state requires that for each repeated experiment a fresh transduction of new cells must be conducted.

Generation of immortalized erythroid cells able to proliferate indefinitely in an undifferentiated state whilst maintaining the capacity to undergo differentiation to generate reticulocytes has been a major goal of the erythroid biology field for decades. Early excitement surrounding the development of induced pluripotent stem cell lines has been tempered by the observation of severe erythroid differentiation defects, expression of fetal globins, and to date minimal capacity for enucleation[10–12]. The capability of orthochromatic erythroblasts, characterized by their condensed nuclei, to support malaria parasite entry[13,14] has led to exploration of cell lines unable to complete differentiation as a model for invasion[15]. For example, a recent study reported invasive susceptibility of semi-differentiated cells of the JK-1 erythroleukemic cell line. These cells display a nucleated polychromatic erythroblast-like morphology and despite supporting parasite invasion were not able to support further parasite development[15]. Although these cells can provide insight into the requirement of receptors, such as basigin, for attachment, and entry[15], the significant membrane complex remodeling and reduction of membrane protein abundance (basigin and CD44 in particular) that occur prior to and during erythroblast enucleation[7,16,17] means that observations made using this model may not extrapolate well to anucleate red blood cells.

In 2017, Trakarnsanga et al.[18] reported the generation of the first immortalized human adult erythroblast cell line—Bristol Erythroid Line Adult (BEL-A). Able to proliferate indefinitely as undifferentiated proerythroblasts, this line can be induced to undergo differentiation and enucleation, generating reticulocytes that are functionally identical to those derived from primary cell cultures. Expanding BEL-A cells can be lentivirally transduced with high efficiency, and are amenable to CRISPR-Cas9-mediated gene editing for the generation of stable clonal cell lines with knockout of individual and even multiple blood groups[19]. In addition, reticulocytes generated through in vitro differentiation of the BEL-A cell line, whether unedited or as edited sublines, derive from the same donor, eliminating the impact of donor variability and polymorphisms between experiments.

In this study, we exploit differentiation of the BEL-A cell line to generate enucleated reticulocytes that can sustain invasion and growth of *P. falciparum*. By employing CRISPR-Cas9-mediated receptor gene knockout and lentiviral expression of wild-type and mutant genes for complementation of invasion defects, we present a sustainable model system that allows us to interrogate the requirement for specific domains and associations of essential receptor complexes during the process of *P. falciparum* invasion.

## Results

**P. falciparum invasion and development in BEL-A reticulocytes.** In order to determine whether BEL-A-derived reticulocytes represent a suitable model for the study of malaria parasite invasion, expanding BEL-A cells were induced to undergo terminal erythroid differentiation. After 15 days, enucleated reticulocytes were purified by leukofiltration under gravity and subjected to invasion assays, in which magnet purified *P. falciparum* schizonts were added to target cells. Reticulocytes derived from the BEL-A cell line in this manner were susceptible to invasion, with robust parasitemia equivalent to that observed in parallel using donor red blood cells as noted by the appearance of ring stage (immature) parasites 2 h post-parasite incubation (Fig. 1a, b). To assess the efficiency of invasion into BEL-A-derived reticulocytes, we manually evaluated Giemsa-stained cytospins and calculated the parasite multiplication rate (PMR), or the ratio of resultant ring stage parasites to mature schizonts added. The PMR was statistically indistinguishable for invasion into BEL-A-derived reticulocytes, red blood cells, and primary CD34$^+$ cell-derived reticulocytes in our assays (Fig. 1c), indicating that parasites can invade all three cell types at the same rate. The PMR value (0.6) is lower than reported PMRs[15], which we attribute to the low hematocrit at which these assays were performed. Interestingly, the selectivity index, which measures the propensity of the cells to support multiple invasions, was approximately twofold higher in reticulocytes derived from either cell source compared with red blood cells (Fig. 1d and Supplementary Fig. 1). To assess whether rings observed in the reticulocytes complete the intracellular replication cycle, cytospins of infected BEL-A-derived reticulocytes and red blood cells were taken at several timepoints post-invasion across the intra-erythrocytic cycle (Fig. 1a, b). Representative images of trophozoites, schizonts, and new rings are presented, confirming the capacity of BEL-A-derived reticulocytes to undergo growth and reinvasion with equivalent rates of intracellular parasite development between cell types.

**Flow cytometric quantification of reticulocyte invasion.** The ability to measure parasitemia and calculate invasion efficiency

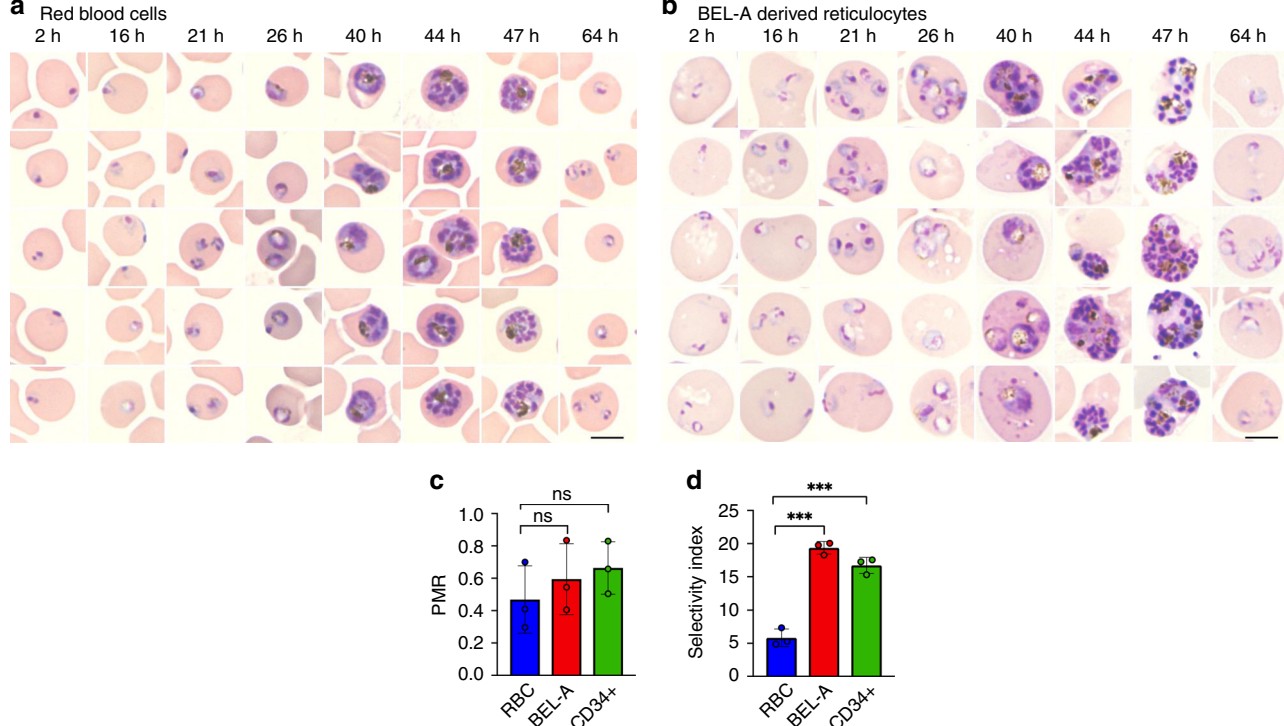

**Fig. 1** BEL-A-derived reticulocytes support invasion and development of *Plasmodium falciparum*. **a** Representative images of Giemsa-stained cytospins depicting *P. falciparum* D10 ring stage parasites following successful invasion of donor erythrocytes or **b** BEL-A-derived reticulocytes at indicated timepoints illustrate development of trophozoites, schizonts at equivalent rates with appearance of new rings indicating reinvasion. Black scale bars, length 5 µm, are shown (bottom right). **c** Bar graph demonstrating parasite multiplication rate (PMR), or the ratio of ring stage parasites to added mature schizonts, for BEL-A and CD34+ cell-derived reticulocytes compared with donor red blood cells. Invasion was quantified through manual counting of rings at 16 h on assessment of Giemsa-stained cytospins. **d** Bar graph demonstrating Selectivity Index of BEL-A and CD34+-derived reticulocytes and red blood cells. The data shown in **c**-**d** are the mean and standard deviation of three biological replicates ($n = 3$), and individual data points are represented as filled circles. A two-tailed *t* test was used to calculate *p* values, where *** indicates $p \leq 0.001$ and ns indicates $p > 0.05$. Source data are provided as a Source Data file

using dyes that stain nucleic acids combined with flow cytometry provides a rapid and high-throughput alternative to manual counting of Giemsa-stained smears or cytospins. As application of nucleic acid dyes for this purpose is reliant upon the absence of DNA in uninfected red blood cells, use of nucleated erythroid cells as a model for invasion precludes their use. In vitro differentiation of primary CD34+ cell-derived erythroblasts or BEL-A cells generate enucleated reticulocytes that can be purified by leukofiltration, which removes nucleated precursors and extruded pyrenocytes. To determine whether flow cytometry could be used to measure invasion efficiency in BEL-A-derived reticulocytes, target cells incubated with schizonts for 16 h were stained with SYBR green for 20 min and analyzed by flow cytometry. Although background staining of reticulocytes (derived from either CD34+ or BEL-A cells) is greater than that of erythrocytes (attributable to SYBR green staining of reticulocyte RNA), invaded reticulocytes could be distinguished, and invasion efficiency quantified at different multiplicities of infection for single and multiply invaded cells using this approach (Fig. 2, Supplementary Fig. 2). Flow cytometry yielded values comparable to manual counting for the PMR and selectivity index (Supplementary Fig. 3).

**Receptor knockout and complementation in BEL-A reticulocytes.** Since the immortalized nature of the BEL-A cell line enables use of CRISPR-Cas9-mediated gene editing for knockout of individual or multiple proteins and clonal selection, we next sought to validate use of CRISPR-Cas9 editing in the context of invasion studies. Cells were transduced with a lentiviral vector co-

expressing Cas9 and a guide targeting the *BSG* gene. Transduced cells were puromycin selected and a population in excess of 80% was found to display a null BSG phenotype based on flow cytometric assessment with the monoclonal antibody HIM6. Individual null cells were fluorescence-activated cell sorting (FACS)-sorted on this basis into 96-well plates, expanded and the null phenotype verified by flow cytometry and immunoblotting. Compound heterozygous mutations within the vicinity of the guide site were confirmed by Sanger sequencing with subsequent ICE (Influence of CRISPR Edits) software analysis (Supplementary Fig. 4)[20]. BEL-A cells expanded from the selected clone were differentiated to verify capacity for enucleation; complete absence of basigin on reticulocytes was confirmed by flow cytometry (Fig. 3a), whereas expression of other known malaria receptors GPA, GPC, band 3, CD55, and CD44 was unaffected as compared with unedited BEL-A-derived reticulocytes (Fig. 3b).

Whilst previous use of in vitro-derived reticulocytes has focused exclusively upon depleting expression of candidate receptors, complementation or rescue studies for the reintroduction of modified proteins using such a system have not been reported. To assess the capacity to successfully rescue the predicted invasion defect brought about by absence of basigin, sequences encoding full-length wild-type BSG sequence (containing silent mutations within the BSG gRNA site) were cloned into the lentiviral vector pLVX-Neo for expression in BSG KO BEL-A cells such that it was resistant to editing by the Cas9 and gRNA constitutively expressed in these cells.

Transmission of extracellular receptor binding events to downstream intracellular processes by the cytoplasmic domain

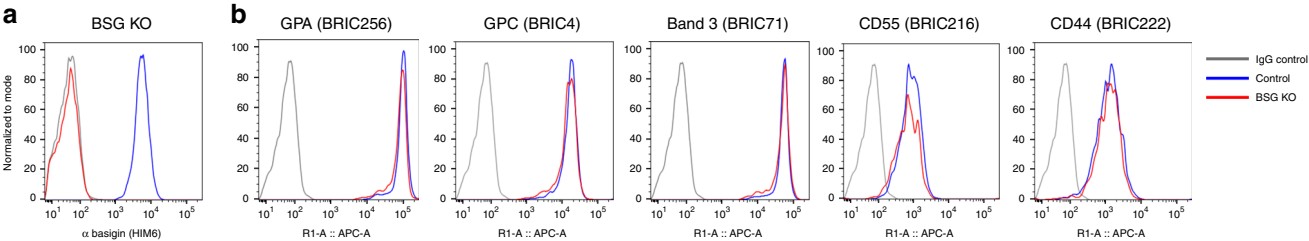

**Fig. 2** Flow cytometry can be used to quantify invasion in BEL-A-derived reticulocytes. Flow cytometry dot plots illustrate ability to quantify invasion using flow cytometry by identification of uninvaded, singly, and multiply-invaded populations of red blood cells, BEL-A, or CD34+ cell-derived reticulocytes based on SYBR Green labeling intensity. Source data are provided as a Source Data file

**Fig. 3** CRISPR-mediated gene editing of BEL-A cells enables generation of BSG knockout reticulocytes. **a** Flow cytometry histogram illustrates basigin (HIM6) labeling in reticulocytes derived from unedited (blue) and basigin knockout (red) BEL-A cell lines compared with IgG isotype control (gray). **b** Flow cytometry histograms illustrate unaltered expression of indicated host receptors between reticulocytes derived from unedited and basigin knockout BEL-A cells. Source data are provided as a Source Data file

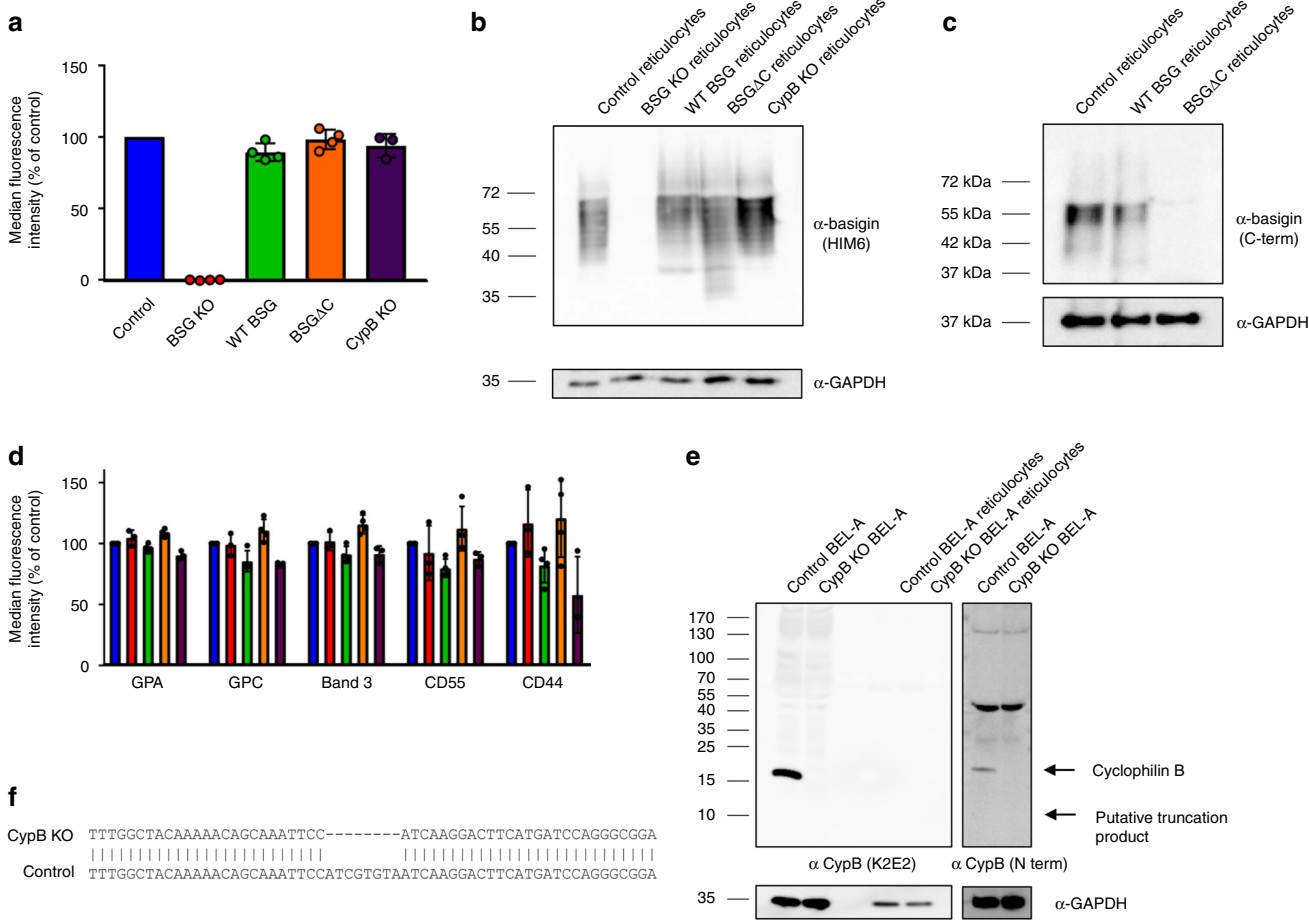

**Fig. 4** Lentiviral expression of exogenous basigin in BSG KO BEL-A cells allows for efficient rescue of receptor presentation in reticulocytes. **a** Bar graphs illustrating expression of basigin as assessed by flow cytometry on reticulocytes derived from indicated cell lines. Data are normalized to endogenous expression of basigin in reticulocytes derived from unedited BEL-A cells and depict the average median fluorescent intensity across four independent cultures ($n = 4$) or 3 ($n = 3$) in the case of CypB KO. Error bars represent standard deviation of the mean, and individual data points are represented as filled circles. **b** Immunoblots of basigin and GAPDH in reticulocytes derived from indicated BEL-A cell lines. **c** Immunoblot of basigin using C-terminal basigin antibody in reticulocytes derived from indicated BEL-A cell lines. **d** Bar graphs illustrating expression of reported malaria receptors on reticulocytes derived from indicated cell lines. Data are normalized to endogenous expression of each receptor in reticulocytes derived from unedited BEL-A cells and depict the average median fluorescent intensity across three or four independent cultures ($n = 3$ for BSG and CypB KO, $n = 4$ for WT and BSGDC). Error bars represent standard deviation of the mean, and individual data points are represented as black circles. **e** Immunoblots of lysates from undifferentiated control (unedited) and *PPIB* (CypB) knockout BEL-A cells and BEL-A-derived reticulocytes with anti-cyclophilin B and anti-GAPDH antibodies demonstrate absence of cyclophilin B (and putative truncation product) expression in *PPIB* CRISPR gene edited cells. **f** Sanger sequencing of *PPIB* gene in edited cells reveals a homozygous 8 base pair deletion at position 284 illustrated by sequence alignment that results in a frameshift. Source data are provided as a Source Data file

of integral membrane proteins is a common theme across biology and whilst the role of the basigin cytoplasmic domain in red blood cells has not been defined, signaling functions of the C-terminus have been previously reported in other cell types[21–23]. Therefore, to explore the requirement for the cytoplasmic domain of basigin for successful invasion, a similar construct was generated in which the BSG cytoplasmic tail was truncated (from 40 to 5 residues).

BSG KO BEL-A cells were transduced with lentiviral vectors for expression of the wild-type (WT) or truncated (BSGΔC) basigin, resulting in populations with mixed surface presentation of the BSG extracellular domain. To maximize the possibility of obtaining reticulocytes in which expression of the reintroduced BSG was equivalent to that endogenously expressed in unedited reticulocytes, transduced undifferentiated BEL-A cells were labeled with anti-basigin HIM6 and individual clones FACS-sorted to match BSG expression in modified BEL-A cell lines to that of untransduced cells (Supplementary Fig. 5). In each case,

selected clones were induced to undergo differentiation, with capacity for enucleation verified and expression of BSG as well as other receptors assessed. Figure 4a shows 89.3 ± 5.2% (standard deviation) rescue of reticulocyte surface expression upon reintroduction of WT BSG with 98.3 ± 4.2% for BSGΔC. Complete absence of basigin expression in BSG KO reticulocytes and altered electrophoretic mobility of BSGΔC was confirmed by immunoblotting (Fig. 4b). Immunoblotting with an antibody specific to the C-terminus of basigin enables detection of full-length but not truncated protein as expected, confirming absence of the cytoplasmic domain in BSGΔC reticulocytes (Fig. 4c). Expression of other parasite-associated erythrocyte surface receptors was not substantially altered (Fig. 4d and Supplementary Fig. 6).

The immunophilin protein cyclophilin B was recently reported to associate with basigin to form a host multiprotein receptor complex that may be required for invasion[24]. No null erythroid phenotypes have been reported for cyclophilin B. Therefore, to

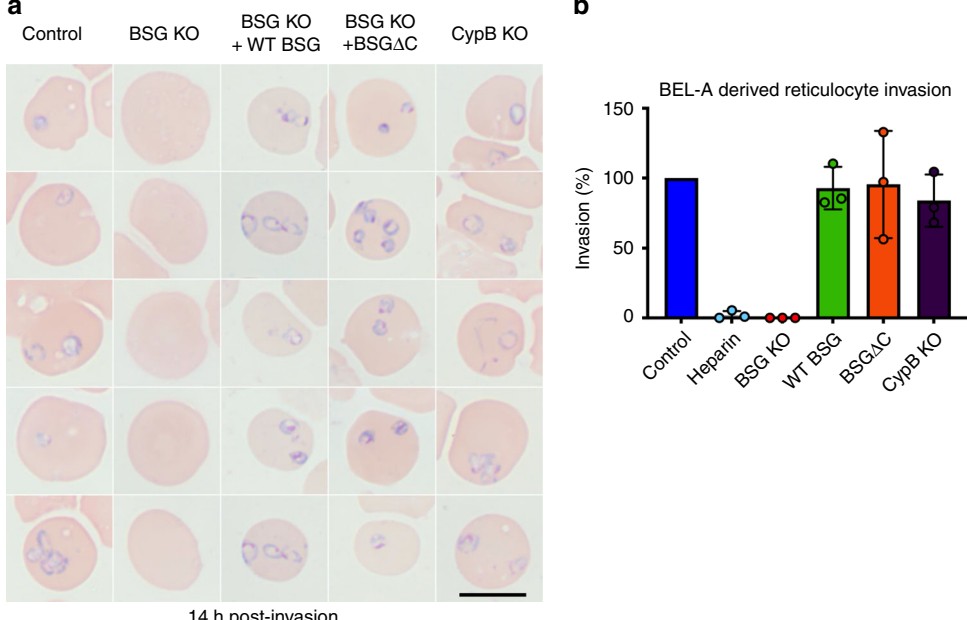

**Fig. 5** Basigin-dependent *Plasmodium falciparum* invasion of reticulocytes can be ablated and complemented through genetic manipulation of BEL-A cells. **a** Rings are completely absent in reticulocytes derived from BSG KO BEL-A cells but are observed in reticulocytes derived from WT BSG and BSGΔC rescue lines and from a CypB KO. Black scale bar shown at bottom right is 10 μm. **b** Bar graph illustrating quantification of invasion of reticulocytes derived from indicated lines. Data represent invasion efficiency normalized to invasion in unedited control BEL-A-derived reticulocytes and assessed through blinded manual counting of Giemsa-stained cytospins from three independent experiments ($n = 3$). Error bars represent standard deviation of the mean, and individual data points are represented as filled circles. A one sample $t$ test showed no significant difference in invasion into control BEL-A cells compared with invasion into WT BSG rescue, BSGΔC rescue, and CypB KO lines. Assays in which heparin is used to inhibit invasion provide a negative control. Source data are provided as a Source Data file

generate novel CypB KO reticulocytes and confirm its involvement in basigin-mediated parasite invasion, BEL-A were transduced with pLentiCRISPRv2 containing a guide targeting the *PPIB* (CypB) gene. Expression of CypB was undetectable by flow cytometry in either undifferentiated BEL-A cells or red blood cells (Supplementary Fig. 7). CypB was also undetectable in BEL-A-derived reticulocytes as measured by immunoblotting; however, a high level of expression was observed in undifferentiated BEL-A cells (Fig. 4e) consistent with its reported downregulation during erythropoiesis[25] (Supplementary Fig. 8). Transduced cells were puromycin selected and blind-sorted to derive single clones. Knockout of CypB was confirmed by immunoblotting of edited and unedited undifferentiated BEL-A cell lysates, confirming complete absence of detectable full-length protein or putative truncation product (Fig. 4e). Sanger sequencing identified a homozygous 8 bp deletion at position 284 resulting in frameshift (Fig. 4f). Reticulocytes derived from CypB KO BEL-A cells were found to express normal levels of BSG and other malaria receptors with the exception of a mild reduction in GPC (Fig. 4d).

**Complementation of basigin knockout restores invasion**. To assess invasive susceptibility of modified reticulocytes, *P. falciparum* schizonts were magnetically purified and added to $5 \times 10^5$ target cells in a 96-well plate. After 16 h, cytospins were prepared, Giemsa-stained, and invasion quantified by manual counting of rings. Figure 5a (left) illustrates the anticipated complete absence of invasion in BSG KO reticulocytes in agreement with previous studies[2,15] and quantified in Fig. 5b. An invasion assay in which unmodified and BSG KO BEL-A-derived reticulocytes were incubated with schizonts at high multiplicity of infection resulted in ~ 60% parasitemia in unedited cells, with no rings observed in BSG KO reticulocytes, confirming the phenotype even under extreme invasive pressure (Supplementary Fig. 9). Of note, no

significant difference in invasive susceptibility was observed between unmodified reticulocytes and reticulocytes derived from the *PPIB* (CypB) knockout line across three independent experiments encompassing assays at both low and high multiplicity of infection and rates of invasion (Fig. 4b, Supplementary Fig. 10).

Reintroduction of WT BSG on an endogenous BSG KO background results in the complete restoration of invasive susceptibility (Fig. 5a, b). Strikingly, expression of C-terminally truncated basigin also rescued invasive susceptibility to levels equivalent to that of unmodified BEL-A-derived reticulocytes. These data demonstrate for the first time the capacity for complementation of invasion defects through genetic manipulation of in vitro-derived reticulocytes.

## Discussion

The ability to disrupt and functionally complement phenotypes through genetic knockout and exogenous gene expression is a cornerstone approach in genetics and cell biology; however, direct application of these techniques to red blood cells is precluded by their anucleate nature. The development of systems for the in vitro culture of nucleated erythroid precursors that can be manipulated prior to their differentiation to enucleated reticulocytes, however, has paved the way to their application within the field of red blood cell biology.

Using enucleated cells derived through in vitro differentiation of the recently described immortalized adult erythroblast cell line, BEL-A, we demonstrate the capacity of these reticulocytes to support invasion by and growth of the malaria parasite *P. falciparum*. Using CRISPR-mediated knockout of the gene encoding the essential host receptor basigin in BEL-As we report the generation of basigin null reticulocytes. These reticulocytes are completely refractory to invasion by *P. falciparum*, confirming

essentiality of this receptor for invasion[2]. By lentiviral introduction of exogenous CRISPR-resistant wild-type basigin on an endogenous basigin knockout background, we generate reticulocytes with close to endogenous levels of this receptor and observe complete rescue of invasive susceptibility in reticulocytes derived from this line, demonstrating the ability to genetically complement a *P. falciparum* invasion defect that results from absence of an essential red blood cell host receptor.

Although these data firmly establish the role of the extracellular domain of basigin as a site for binding of the merozoite PfRh5, the molecular consequences of this binding event within the host cell remains controversial, with proposed consequences including the formation of an opening or pore between parasite and host, and a $Ca^{2+}$ influx[26–28]. The role of the C-terminal cytoplasmic domain of basigin is poorly understood; it has been shown to have a signal inhibitory function at the T-cell synapse[23] and reduces the sensitivity of intracellular store-operated $Ca^{2+}$ to cGMP in hepatoma cells;[21,22] however, there have been no studies describing its function in red blood cells. To explore the hypothesis that the cytoplasmic domain of basigin participates in merozoite invasion in response to PfRh5 binding of the extracellular domain, a C-terminally truncated mutant was expressed in basigin knockout BEL-A cells. Reticulocytes differentiated from this modified cell line were found to exhibit no significant difference in susceptibility to invasion by *P. falciparum*. This excludes a requirement for the cytoplasmic domain in parasite invasion.

In addition to enabling dissection of the molecular basis of host receptors with established involvement in invasion, the ability to generate novel reticulocyte phenotypes with complete deficiency of prospective receptors provides a model for the verification or exclusion of candidate receptors identified via less direct approaches. A recent report proposed the existence of a multi-protein complex between basigin and cyclophilin B within the erythrocyte membrane, with a synthetic cyclophilin B-binding peptide shown to inhibit merozoite invasion, leading to the proposal that cyclophilin B is a host receptor for *P. falciparum*[24]. As there have been no reports of cyclophilin B knockout red blood cells that would enable direct assessment of this hypothesis, we generated a cyclophilin B knockout BEL-A cell line, from which reticulocytes were derived. In contrast to the complete ablation of invasion in basigin knockout reticulocytes, no significant difference in invasive susceptibility was observed in cyclophilin B knockout reticulocytes compared with unedited controls. Through generation of this novel phenotype, we thus conclude that host cell-derived cyclophilin B is not a crucial receptor for merozoite invasion of the red blood cell.

In demonstrating the capacity for intracellular development and reinvasion of BEL-A-derived reticulocytes, this work opens the door to potential future development of continuous growth assays over multiple cycles; however, we note that the economic costs associated with generating sufficient reticulocytes for such experiments at scale are currently prohibitive. Optimization of reticulocyte yield, culture costs, and conditions for extended storage of reticulocytes should represent areas of future development in this context. Interestingly, although a similar PMR was observed between mature red blood cells and reticulocytes derived from either primary CD34+ or BEL-A cell-derived reticulocytes, reticulocytes derived from either cell source demonstrated a two to threefold greater propensity to support multiple invasion events. This may reflect the increased surface area of the larger reticulocyte for merozoite attachment, or indicate a reticulocyte membrane-cytoskeletal architecture more permissive to invasion than that of mature erythrocytes.

The amenability of the BEL-A cell line to genetic manipulation (including via gene editing), coupled with the ability of BEL-A-derived reticulocytes to permit the entire red blood cell development cycle of the parasite, allows for wide-ranging manipulation of receptors and other host proteins involved in parasite invasion, development, and egress.

Whilst the capacity for edited erythroid cells to undergo successful differentiation and enucleation must always be considered, future studies will undoubtedly exploit advances in the application of CRISPR guide libraries for generating panels of sustainable receptor knockout lines for the study of a variety of invasion-associated phenotypes as well as alternative editing approaches for site specific modification of endogenously expressed host proteins. Notably, BEL-A-derived reticulocytes express both the Duffy blood group protein and transferrin receptor[19] and thus should be susceptible to invasion by other malaria species including *Plasmodium vivax* and *Plasmodium knowlesi*.

In summary, we present here data that establish reticulocytes derived through differentiation of the immortalized erythroblast cell line BEL-A as a robust model system for the exploration of host protein involvement in malaria invasion. We provide evidence that *P. falciparum* merozoites are able to invade and undertake the complete intracellular development cycle within the reticulocytes derived from this line. Further, using CRISPR-mediated gene knockout we demonstrate capacity to generate novel reticulocyte receptor knockout phenotypes, recapitulating known invasion defects and challenging indirect evidence in support of others. Through lentiviral expression of wild-type and truncated basigin on a background of endogenous protein knockout, we additionally present the first demonstration of complementation of a receptor-associated invasion defect whilst excluding a role for the cytoplasmic domain of basigin during this process. Overall, these data establish a model system that will enable detailed dissection of host protein involvement in multiple aspects of malaria parasite pathology.

## Methods

**Cloning.** Lentiviral vector pLentiCRISPRv2 containing guide sequence 5′-TTC ACTACCGTAGAAGACCT-3′ targeting BSG or 5′-TGAAGTCCTTGATTACA CGA-3′ targeting *PPIB* (CypB) was ordered from Genscript. Lentiviral expression constructs for complementation experiments were generated using a pLVX-Tight-Puro plasmid modified to contain a CMV enhancer and promoter and a neomycin resistance gene. Gibson assembly was used to combine the NotI-linearized plasmid with sequence encoding either full-length BSG, or BSG lacking the cytoplasmic domain (residues 1–234; BSGΔC). The human BSG gene was codon re-optimized, including six mutations in the CRISPR guide sequence used to knockout BSG (new sequence 5′-TTTACCACCGTGGAGGATCTGG-3′). The re-optimized gene was synthesized commercially with flanking sequences (5′ flank CTAGCGCTACCGG TCGCCACCGGATCCACC; 3′ flank 5′-GCGGCCGCGCCGGCTCTAGATCG CG-3′) for Gibson assembly to generate the full-length BSG construct. To generate the BSGΔC construct, PCR, using the full-length synthetic gene as a template, was used to add sequences for Gibson assembly (primers 5′-CTAGCGCTACCGGTCG CCACCGGATCCACCATGGCCGCCGCCCTCTTTGTC-3′ and 5′-CGCGATCT AGAGCCGGCGCGGCCGCTCACTTCCGCCGCTTCTCGTAGATG-3′).

**BEL-A cell culture.** BEL-A (Bristol Erythroid Line–Adult) cells were cultured as per the method originally described by Trakarnsanga et al.[18] and expanded on by Hawksworth et al.[19]. In the expansion phase, cells were cultured at a density of $1–3 \times 10^5$ cells/ml in expansion medium, which consisted of StemSpan SFEM (Stem Cell Technologies) supplemented with 50 ng/ml SCF, 3 U/ml erythropoietin, 1 μM dexamethasone (Sigma-Aldrich) and 1 μg/ml doxycycline (Takara Bio). Complete medium changes were performed every 48 h. In the differentiation phase, cells were seeded at $2 \times 10^5$/ml in differentiation medium (Iscove's Modified Dulbecco's Medium (IMDM; Source BioScience UK Ltd), supplemented with 3 U/ml erythropoietin (Bristol Royal Infirmary), 3 U/ml heparin (Sigma), 0.5 mg/ml holotransferrin (Sigma), 3% v/v heat-deactivated Human Male AB Serum (Sigma), 2% (v/v) fetal calf serum (Hyclone), 10 μg/ml insulin (Sigma), 100 U/ml penicillin (Sigma) and 100 μg/ml streptomycin (Sigma), 1 ng/ml IL-3, 40 ng/ml SCF and 1 μg/ml doxycycline). After 2 days, cells were reseeded at $3.5 \times 10^5$/ml in differentiation medium. On differentiation day 4, cells were reseeded at $5 \times 10^5$/ml in fresh differentiation medium without doxycycline. On differentiation day 6, a complete media change was performed, and cells were reseeded at $1 \times 10^6$/ml. On day 8, cells were transferred to tertiary culture medium (IMDM supplemented as

above in the absence of SCF and IL-3) and maintained at $1 \times 10^6$/ml with complete medium changes every 2 days until day 14.

**CD34$^+$ cell culture**. For isolation of peripheral blood mononuclear cells (PBMCs), the blood sample was mixed with 0.6% v/v citrate-dextrose solution (ACD; Sigma), diluted 1:1 with Hanks balanced salt solution (HBSS; Sigma) with 0.6% v/v ACD and layered on top of 25 ml Histopaque 1077 (Sigma). The sample was then centrifuged at 400 $g$, at room temperature (RT) for 35 min. The interface layer consisting of density-purified mononuclear cells was then collected, washed three times in HBSS and resuspended in 12 ml cold Red Cell Lysis Buffer (NH$_4$Cl, 4.15 g/L; EDTA, 0.02 g/L; KHCO$_3$, 0.5 g/L) at 4 °C for 10 min, to provoke lysis of any remaining erythrocytes. Cells were washed twice in HBSS and counted in a haemocytometer using the Trypan Blue dye (Sigma) exclusion test to distinguish between dead and live cells.

CD34$^+$ magnetic cell isolation was performed on the PBMCs according to the manufacturer's protocol for the Direct CD34$^+$ progenitor cell isolation kit (Miltenyi Biotec), to enrich for haematopoietic progenitor cells. Cells were cultured according to a protocol initially described by Griffiths et al[7]. Isolated cells were counted and plated at a density of $1 \times 10^5$ cells/ml in a primary expansion medium. This primary medium was IMDM (Source BioScience UK Ltd) supplemented with 3U/ml erythropoietin (Bristol Royal Infirmary), 3U/ml heparin (Sigma), 0.5 mg/ml holotransferrin (Sigma), 3% v/v heat-inactivated Human Male AB Serum (Sigma), 2 mg/ml Human Serum Albumin (HSA; Sigma), 10 µg/ml insulin (Sigma), 100 U/ml penicillin (Sigma) and 100 µg/ml streptomycin (Sigma), with extra supplementation of 40 ng/ml Stem Cell Factor (SCF; Miltenyi Biotec) and 1 ng/ml IL-3 (R&D Systems) to induce cell proliferation. The cells were incubated at 37 °C in 5% CO$_2$ in this primary medium with daily media addition from Day 3 to Day 7 of culture. From Day 8 to Day 12, secondary medium was added instead, which consisted of the same IMDM base supplemented with 40 ng/ml SCF. After Day 13, tertiary medium consisting of the IMDM base without growth factor additions was used in order to induce terminal erythroid differentiation. On Day 21, when the cells typically achieved a density in culture of $2 \times 10^6$ cells/ml, reticulocytes were purified through leukofiltration of the culture to remove nuclei and nucleated cells.

**Leukofiltration**. For leukofiltration, a leukocyte reduction filter (NHSBT, Filton, Bristol) was pre-soaked and equilibrated with phosphate-buffered saline (PBS) and the cultured cell suspension was loaded into the filter followed by at least three volumes of PBSAG and allowed to pass through under gravity. The resulting flow-through was then centrifuged at $400 \times g$, for 20 min and the pelleted cells were resuspended in PBSAG. The purified reticulocytes were then stored at 4 °C.

**Lentiviral transduction**. HEK293T cells (Clontech) were cultured in Dulbecco's Modified Eagle Medium (DMEM) (Gibco) containing 10% fetal calf serum (Gibco). Cells were seeded in 10 cm dishes and calcium phosphate transfected using lentiviral packaging vectors pMD2 (5 µg) and pPAX (15 µg) and the lentiviral vector of interest (20 µg). After 24 h, DMEM was removed and replaced with 5 ml fresh media. Virus was harvested after 48 h, concentrated using Lenti-X concentrator (Clontech) according to the manufacturers protocol and stored at − 80 °C. Concentrated virus equivalent to that harvested from half a 10 cm dish of HEK293T cells was added to $2 \times 10^5$ BEL-A cells in the presence of 8 µg/mL polybrene (Sigma) for 24 h. Cells were subsequently washed three times in PBS and resuspended in fresh media. For vlentiCRISPRv2 transductions cells were selected 24 h after removal of virus using 1 µg/ml puromycin for 48 h.

**Selection of individual clones by FACS**. BEL-A cells transduced with plenti-CRISPRv2 containing guide targeting BSG were immunolabelled with propidium iodide and anti-basigin antibody HIM6.Individual cells within the negative population FACS were sorted into a 96-well plate for onward culture using a BD Influx Cell Sorter. To derive clones of BSG knockout cells transduced with pLVX constructs in which basigin expression was matched to endogenous levels, transduced populations were immunolabelled with HIM6 and single clones FACS isolated through matching to a tight gate based on endogenous basigin expression of unedited BEL-A cells. Derivation of *PPIB* (CypB) knockout clones was achieved through blind sorting of individual clones followed by downstream screening using Sanger sequencing and immunoblotting.

**Flow cytometry**. For flow cytometry on undifferentiated BEL-As, $1 \times 10^5$ cells resuspended in PBSAG (PBS + 1 mg/ml BSA, 2 mg/ml glucose) + 1% BSA were labeled with primary antibody for 30 min at 4 °C. Cells were washed in PBSAG, incubated for 30 min at 4 °C with appropriate APC-conjugated secondary antibody, and washed and data acquired on a MacsQuant VYB Analyser using a plate reader. For differentiated BEL-As, cells were stained with 5 µg/ml Hoechst 33342 then fixed (if required) in 1% paraformaldehyde, 0.0075% glutaraldehyde to reduce antibody binding-induced agglutination before labeling with antibodies as described. Reticulocytes were identified by gating upon Hoechst-negative population.

**Sequencing of CRISPR-edited BEL-A clones**. For verification of CRISPR edits, genomic DNA was isolated from specific clones using a DNeasy Blood and Tissue

Kit (Qiagen). DNA regions encompassing guide sites were amplified using primers specific for basigin: FWD 5′-TGAAAGCAGGAAGGAAGAAATG-3′ REV 3′-TCAAACCCTGGGACTTCAC-5′ and PPIB FWD 5′-GCCCGCTCACTTAGTAGCAC-3′, REV 3′-ATCGCGTACCCACATGTCTT-5′. In each case the forward primer was used for Sanger sequencing performed by Eurofins MWG.

**Antibodies**. Mouse monoclonal antibodies used were as follows: BRIC4 (GPC), BRIC216 (CD55), BRIC222 (CD44), BRIC71 (band 3), BRIC256 (GPA) (all IBGRL hybridoma supernatants used 1:2), HIM6 (basigin) (Biolegend [1:50 flow cytometry, 1:500 immunoblotting]), ab64616 (basigin C-terminal) (AbCam, 1:500)), K2E2 (CypB) (Santa Cruz, [1:50 flow cytometry, 1:500 immunoblotting]), SAB2101856 (CypB N-terminal) (Sigma 1:500), GAPDH 0411 (Santa Cruz) (1:1000), IgG1 control MG1-45 (1:50 Biolegend). Secondary antibodies were allophycocyanin (APC)-conjugated monoclonal rat anti-mouse IgG1 RMG1–1 (Biolegend 1:50), swine anti-rabbit HRP (P0399), or rabbit anti-mouse HRP (P0260) (Dako 1:2000).

**Parasite culturing**. *P. falciparum* parasites of a D10 derived parasite strain (D10-PHG)[29] were maintained in human erythrocytes at between 2 and 4% hematocrit using standard culture conditions[30]. The culture medium consisted of Roswell Park Memorial Institute 1640 containing 5.96 g/L HEPES, 2 g/L sodium bicarbonate, and 0.0053 g/L Phenol Red (Sigma), supplemented with 0.05 g/L hypoxanthine (Sigma), 0.025 g/L gentamycin (Sigma), 0.3 g/L L-glutamine (Sigma), and 5 g/L AlbuMAX II (Thermo Fisher Scientific). Cultures were incubated at 37 °C in a gas mixture of 5% O$_2$, 5% CO$_2$, 90% N$_2$.

**Invasion assays into erythrocytes and BEL-A-derived reticulocytes**. Schizont stage parasites were magnetically purified using the Magnetic Cell Separation (MACS) system (Miltenyi Biotec)[31] and added to wells of a 96-well plate containing either erythrocytes or leukofiltered BEL-A-derived reticulocytes in culture medium. Each well contained $5 \times 10^5$ cells, with cell numbers counted using a hemocytometer, in a final volume of 200 µl. Heparin (100 mU/µl final) was used to inhibit invasion in negative controls. After ~ 16 h, invasion was quantified using cell counting and flow cytometry. For cell counting, $1.5 \times 10^5$ cells were applied to a slide using a cytocentrifuge. Slides were immersed in 100% methanol fixative (15 min), Giemsa stain (10 min), and water (3 min), and imaged using a Leica DMR microscope fitted with a Zeiss AxioCam HR camera. For quantification of invasion at least 1000 cells were counted per cytospin.

For flow cytometry, cells were washed in PBSAG, stained with SYBR Green (1:1000 in PBS; Sigma-Aldrich) for 20 min at room temperature in the dark, and washed three times in PBSAG. In total, $1 \times 10^5$ cells from each well were acquired using the fluorescein isothiocyanate channel of a BD Fortessa flow cytometer.

**Reporting summary**. Further information on research design is available in the Nature Research Reporting Summary linked to this article.

## Data availability

The data that support the findings of this study are available from the corresponding authors upon reasonable request. The source data underlying Figs. 1c–d, 4a–e, and 5b, and Supplementary Figs. 2–4, 9, and 10 are provided as a Source Data File.

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

## Acknowledgements

The authors wish to acknowledge the assistance of Dr. Andrew Herman and Lorena Sueiro Ballesteros of University of Bristol flow cytometry facility for cell sorting. This research was funded by a National Institute for Health (NIHR) grant to support a NIHR Research Blood and Transplant Unit (NIHR BTRU) in Red Blood Cell Products at the University of Bristol in Partnership with NHSBT (NIHR-BTRU-2015-10032; T.J.S., J.F., A.M.T.), grant funding from NHS Blood and Transplant R&D committee (NHSBT WT15-05; T.J.S., K.L.H.-S., A.M.T.) and Wellcome (Investigator Award 100993/Z/13/Z to J.B.). K.E.W. is supported through a Henry Wellcome Postdoctoral Fellowship (107366/Z/15/Z), P.L.M. was funded by the European Union (F.A. H2020-MSCA-ITN-2015, "RELEVANCE", Grant agreement No. 675117). J.H. was funded by a EPSRC/BBSRC SynBio Centre CDT PhD with Defence Science and Technology Laboratory as an industrial partner. The views expressed are those of the author(s) and not necessarily those of the NHS, the NIHR or the Department of Health and Social Care.

## Author contributions

Experiments were conceived and designed by T.J.S and K.E.W. T.J.S. and K.E.W. carried out the majority of experiments, performed the analysis, and prepared the figures. K.L.H.-S., F.S.-R.T., P.L.M., and J.H. assisted with experimental work, J.F. provided BEL-A cell line. A.M.T. and J.B. contributed equally to the supervision and design of the project and edited the manuscript. T.J.S. and K.E.W. wrote the manuscript. All authors read and approved the manuscript.

## Additional information

**Competing interests:** The authors declare no competing interests.

