## [Peer Review File · Nature Communications]

Reviewers' Comments:

Reviewer #1:

Remarks to the Author:

The manuscript by Satchwell et al. presents use of a recently established immortalised erythroblast cell line, BEL-A, for infection by the human malaria parasite *Plasmodium falciparum*. The authors demonstrate that differentiated (enucleated) BEL-A-derived reticulocytes support *P. falciparum* invasion and intraerythrocytic development. The BEL-A line is further validated for use in the malaria community by demonstrating that genetic knock out of RBC receptors (Basigin) results in a block in parasite invasion. This phenotype can be rescued by complementation/re-introduction. The work described in this paper is promising as the BEL-A line provides a simplified and efficient system for RBC manipulation for the study of parasite phenotypes (and other purposes), compared to the use of erythroleukemic cells or in vitro maturation of hematopoietic stem cells. However, the manuscript does not go beyond validation and as such would be more suitable for a methods journal, unless more new biological insights are provided (including more detailed analysis of the discrepancy between the observed dispensability for CypB and previous work showing the contrary).

Major points

1. The authors provide only a superficial characterisation of *P. falciparum* growth and replication in BEL-A cells. There is no quantitative data on cycle length, PMR etc included. Instead a series of Giemsa smears are shown. To fully assess whether BEL-A cells constitute a suitable model to study host factors in *Plasmodium* development, more quantitative assessment of parasite propagation in these cells would be useful. In particular, addressing following questions would be interesting
 - o Is the parasite multiplication rate (PMR) comparable between BEL-A cells and RBCs? The PMR could be easily calculated by determining the parasitemia before and after the second reinvasion.
 - o How long can parasites be propagated in one batch of differentiated BEL-A cells? This could be addressed by culturing parasites over several cycles and assessing parasitemia/PMR in RBCs and BEL-A cells.
 - o Is the cycle length identical between RBCs and BEL-A cells, or are parasites delayed? From images in Figure 1 it appears that while in RBCs, some segmented schizonts are found at 46 hours, but in BEL-A cells, the schizonts are not yet segmented. Yet, this is obviously difficult to judge from a few selected Giemsa-stained parasites. A more detailed time course analysis could give further insight.
2. Demonstration of CRISPR modification of BEL-A lines as a tool to investigate host factors of *Plasmodium* infection is exciting. However, validation of the method with the basigin knockout is a well-executed proof-of-concept. The authors continue to investigate a lesser-described putative receptor for *Plasmodium*, CypB and found it dispensable for invasion, in contrast to a previous study. Further in-depth analysis addressing the following points could enhance the biological insight of the paper.
 - o The CypB knockout BEL-A line they investigate has a frame shift mutation, potentially leading to the expression of a truncated CypB that is not detected by Western blotting using the selected CypB antibody. While unlikely, there is the minor possibility that the expression of this truncated CypB is sufficient to support invasion of *Plasmodium*. Analysing a second CypB clone with a different mutation would be helpful to further clarify the role of CypB.
 - o The precise parameters under which the invasion experiments were performed are unclear. At which MOI was the infection done, i.e. how many schizonts were added to the 5×10^5 BEL-A cells to obtain the results depicted in Figure 4B? From the way it is written, it appears that two experiments were done, with lower and excess amounts of schizonts. Please show the data for both settings. If the data was obtained using excess schizonts, could it be that a putative mild phenotype of CypB deletion is masked as the invasion capacity is saturated? It might be helpful to show dose-dependent effects using different MOIs.
 - o In the study by Prakash et al., a peptide inhibitor of CypB successfully inhibited *Plasmodium*

invasion. Have the authors tried these inhibitors on their WT and CypB KO BEL-A lines to test if the inhibitory effect of these peptides is specific/due to CypB?

Additional comments

1. Results paragraph 5: "Expression of CypB in reticulocytes was undetectable by immunoblotting, however, a high level of expression was observed in undifferentiated BEL-A cells (Figure 3D)".

According to figure legend and labeling, the blot in Figure 3D does not show any data from differentiated reticulocytes. Instead, it shows the expression of undifferentiated WT BEL-A cells compared to undifferentiated CypB-null BEL-A cells. If the labeling is correct, it should be referred to at the end of the following sentence: "Knockout of CypB was confirmed by immunoblotting of edited and unedited undifferentiated BEL-A cell lysates, confirming complete absence of detectable protein." Please provide an immunoblot demonstrating the absence of CypB in differentiated reticulocytes. Could the authors also comment on why they hypothesize an involvement of CypB in Plasmodium invasion if CypB is not detectable in differentiated reticulocytes?

2. Presentation of data in figure 2 is not in logical order, as initial validation of BEL-1 using the BSG KO is mixed with the subsequent analysis of CypB.

3. Figure 3B: The basigin western blot does not show a clear band, but instead a smear across different molecular weights. While the absence of basigin in the knockout reticulocytes is obvious, it is difficult to see a shift towards a lower molecular weight for the truncated BSG Δ C. It would be helpful to repeat this western blot to achieve a clearer result and higher resolution, and also to provide the expected migration pattern of full-length and truncated basigin.

4. Figure 4C and results, paragraph 8: The depicted flow cytometry plots lack a quantification. In particular the statement in paragraph 8 of the results section "Quantification of invasion efficiency into modified reticulocytes replicated results obtained through manual counting" requires quantitative data of triplicate experiments. Is there a reason that the rightmost population of SybrGreen-positive cells is excluded from the gate? As a control for the background BEL-A cells give in flow cytometry, wouldn't it be possible to subtract the signal of the negative control (Heparin-treated BEL-A cells) from the other experimental conditions? It would also be informative to see how uninfected RBCs and BEL-A cells look like in this flow analysis.

- Introduction, paragraph 2: Bracket missing
- Introduction, paragraph 4: "the significant membrane complex remodeling and loss of protein (basigin and CD44 in particular)". This phrase sounds misleading, suggesting that no basigin is left on the reticulocyte after enucleation, which would contradict the essentiality of basigin for Plasmodium invasion. Clarify that not all basigin is lost during enucleation.
- Results paragraph 3: "To assess the capacity to successfully rescue the invasion defect brought about by the absence of basigin...". This invasion defect has not been shown yet in the paper. The results demonstrating an invasion defect in the BSG KO are presented only in paragraph 6.
- Results paragraph 3: The reasoning for complementing the knockout with a truncated BSG does not become clear. The hypothesis that the cytoplasmic tail is involved in signaling, thus mediating Plasmodium invasion, is only mentioned in the discussion section. Please add a short note here as to why BSG Δ C was investigated.
- Results, paragraph 6: Please refer here already to Figure 4B as it depicts the results of the manual counting of invasion.
- Materials and Methods, Flow Cytometry: In the buffer description of PBSAG, BSA is mentioned twice.
- Figure legend Figure 2: "Flow cytometry histogram illustrates absence of basigin (HIM6) labelling in reticulocytes derived from unedited (blue) and basigin knockout (red) BEL-A cell lines compared to IgG isotype control (grey)." Incorrect description, as basigin labeling is present, not absent, in unedited BEL-A cell lines (as expected). Please correct.

Reviewer #2:

Remarks to the Author:

In their manuscript "Stable knockout and complementation of receptor expression using in vitro cell line derived reticulocytes for dissection of host malaria invasion requirements", Satchwell and colleagues reports the feasibility of using an immortalized erythroblast cell line for differentiating into reticulocytes compatible for invasion of *Plasmodium falciparum*. They also demonstrate potential applications of this system through CRISPR-mediated knock-out of a well known host receptor, basigin leading to invasion inhibition. Authors state several advantages of BEL-A derived reticulocytes for plasmodium research over other existing platforms (such as erythroblasts derived from JK-1 cells), which all appear convincing to this reviewer making it suitable for publication in Nature Communications. This is impactful work, with potential implications in the study of *P. vivax* biology and invasion mechanisms.

Remarks

1. For invasion efficiency comparison (Figure 1), authors matched BEL-A derived reticulocytes to native red blood cells and reports a ratio of 1.08:1 for the number of invasion events. I have to assume that native red blood cells refers to peripheral blood collected from a hospital or blood bank (this information could not be found in the methods section). In such a case, comparison is made between BEL-A derived reticulocytes to mature normocytes which brings some discrepancy. As such, reticulocytes are invaded by most species of plasmodium (including *P. falciparum*) with higher affinity compared to normocytes, with at least a 2-fold higher efficiency for the former. I would therefore recommend the authors to include a comparison between BEL-A derived reticulocytes, normal healthy reticulocytes (purify them from cord blood or peripheral blood using CD47-conjugated magnetic beads or density gradients) and mature normocytes either as part of Figure 1 or as supplemental data.
2. Figure 1: ability of BEL-A derived reticulocytes to support growth and re-invasion of *falciparum*: 38h, 46h and 62h post invasion. Can the authors elaborate on the invasion efficiency in the second cycle compared to the first (62h post invasion) ?. Was the invasion rate higher or lower than the first cycle? During the course of this experiment, does all the non-infected reticulocytes stay as immature reticulocytes or a small fraction mature into more mature reticulocytes (CD71 negative) or even normocytes ? This is subjective, but an immunofluorescence-based assay of CD71 expression as surrogate marker for immature reticulocytes could be adopted.
3. Were there more multiple infections in reticulocytes compared to normocyte controls ?
4. Page 4: Excess purified schizonts were added to 5×10^5 target cells resulted in ~ 60% parasitemia in unedited cells, compare to no rings in BSG KO reticulocytes. What is the excess amount of schizonts to target cells?
5. The authors claims that the flow cytometry method used in this study was capable of rapid initial assessment of invasion in BEL-A derived reticulocytes. However it is best applied only when parasitemia is high. The flow plot clearly shows residual background in BSG KO cells some of which may be due to post-leukofiltration contamination, and un-ruptured schizonts. However, extreme care needs to be taken in drawing conclusions from such experiments as high background from non-infected cells that stains partially positive for SYBR-green. I would still recommend manual counting of Giemsa-stained smears to confirm the invasion efficiency.
6. Flow cytometry plots as the final component of Fig. 4 in some ways deviates the attention of a reader from main conclusions of the paper which is viability of BEL-A reticulocytes for functional

studies in plasmodium. Authors may want to consider re-structuring the sequence of these figures.

Reviewer #3:

Remarks to the Author:

The manuscript by Satchwell et al demonstrates that in vitro differentiation of an enucleation competent immortalized erythroblast cell line (BEL-A) supports the growth of Plasmodium falciparum and that CRISPR/Cas9 editing technology as well as lentiviral expression of transgenes can be used to dissect the role of host receptors in P. falciparum invasion. As such, this provides a very important advance for researchers working on the blood stages of Plasmodium because it now provides a mechanism to determine the role of host proteins in supporting growth and survival of the parasite, which have traditionally relied on much less accessible or more time-consuming approaches (eg. Knockout mice, using donor RBCs from patients that are deficient or have defective host protein of interest, etc). In this study the authors used this system to validate previous findings using several other approaches that basigin is an essential host receptor for Plasmodium invasion and second, that contrary to another study, cyclophilin B is not essential for invasion.

Nevertheless, the study could do with some greater rigor in parts to support their conclusions. Other comments I have provided to improve the flow or clarity of the manuscript.

1. Abstract – The abstract is a bit clunky in parts and could do with rewording to make it clearer. Eg. Using CRISPR-mediated gene knockout and lentiviral expression of open reading frames. This is vague. It would be clearer to say using CRISPR-mediated gene knockout and subsequent complementation of the open reading frame of the gene of interest via lentiviral expression. I would also split up the sentence ‘rescued by re-expression of the receptor’ and ‘mutant thereof’ and introduce the receptor mutant in the following sentence. Eg. Specifically, the ability P. falciparum to invade the edited clone complemented with a basigin mutant lacking its cytoplasmic domain excludes a role for this domain during basigin invasion.
2. The authors have demonstrated that the BEL-A line can support P. falciparum growth. Because this is the first time this is reported it would be useful to provide further insight. How many generations can the parasite grow in the culture, have they looked beyond the cycle after reinvasion (ie. Beyond 62hr)? From Fig 1 it looks like the parasites growing in BEL-A derived reticulocytes have developed more quickly and are more mature at 14 hr and even 38 hr, yet schizonts have not fully-developed by 46 hr when compared to mature RBC. Is the timing of parasite development/stages different between mature RBC and BEL-A derived reticulocytes? Could this be because they are growing in retics rather than more mature RBC or because of the cell line. How many schizonts were added to the cells (ie. What was the ratio?) and what levels of invasion (% parasitemia) were observed?
3. Structure of the article- for the sake of flow, it would have been best after BEL-A BSG null clones were generated to assess at this point the ability of parasites to invade and hence validate that basigin is essential for invasion before doing complementation studies to look at the role of particular domains (ie. At bottom of page 3 of manuscript).
4. 1st paragraph page 4 – The last two sentences are a bit confusing. I would reword to the effect that you cloned full-length and truncated BSG sequences into the lentiviral pLVX-Neo for expression in BEL-A cells such that the guide sites were mutated so that they were resistant to editing.
5. Role of basigin and C-term domain (Figure 3). What is the variation in basigin expression in the

control (Fig 3A)? The western blot with basigin antibody (Figure 3B) is difficult to decipher with so many bands and to me there doesn't appear to be any visible difference between wildtype and the truncation mutant to demonstrate it is indeed truncated. The truncation mutant needs to be confirmed in another manner in order to draw the conclusion that the C-term is not essential for invasion. What about transcript size to demonstrate expected transcript length? In Fig 3C. Is difference in GPC between control and CypB KO statistically significant?

6. Contribution of Cyclophilin B to invasion. That expression of CypB in reticulocytes is undetectable is important when weighing up if this protein is an important receptor for Plasmodium invasion. Presumably it is also absent from mature RBC. It would be good to have a blot or qRT-PCR to look at expression at the different stages. Importantly, the full blot for D should be included to rule out whether a truncated product is still expressed and this should be validated using another approach if the antibodies bind downstream of where the truncation takes place. Why were the FACS plots not performed as per Fig 2 with cyclophilin B antibody in place of basigin antibody?

7. Assessment of invasive susceptibility of modified reticulocytes – how many schizonts were added, what was the ratio of schizont to reticulocytes, and how many cells were manually counted. Fig 4 A should include the unedited control. It would be helpful if Fig 4B showed the SD of % invasion of the control (ie. expressed as 100% but still shows SD).

8. The FACS analysis is important to perform because it is quantitative and can be used to assess a large number of cells to determine whether even very low levels of invasion have occurred in the BEL-A edited lines +/- complementation. Unfortunately the FACS has not been robustly performed. I disagree with the authors that FACS is reliant on the absence of DNA in uninfected RBC and nucleated erythroid cells can be used if a different approach is used. For example, one can discriminate between nucleated uninfected cells and infected reticulocytes or nucleated cells if GFP-expressing parasites are used for invasion, in addition to stains. Presumably the nucleated cells are also much bigger and FSC/SSC could be used to discriminate these as could staining for human equivalent to Ter119/CD44-CD71 as these would differ in maturation stages of erythroid cells. Whilst SYBR green stains reticulocyte RNA, Hoechst could also be used (the nucleated cell most likely gives much higher levels of fluorescence than infected reticulocyte). As it is the authors do see events in the invaded gates but they have to rely on cytopins to say these are nucleated cells and therefore discount them. Also please include the % invaded on the plots in Fig 4C. No mention is made of what the events are that are to the right of the invaded gate? Moreover, what are the 2 peaks in the invade gate?

Discussion.

RhopH3 has been shown in two studies using conditional gene knockout that it is essential for parasite invasion of RBC. Hence the authors should change the last sentence of paragraph 2 page 6 as their work has just shown that cyclophilin B is not essential. It may be that this interaction between RhopH3 and cyclophilin is not a true interaction.

The authors mention that this system could be used to look at parasite development and egress, however, it would be worthwhile pointing out the obvious caveat that knocking out these proteins in the undifferentiated BEL-A line may have dire consequences and may not support differentiation and enucleation.

Other comments:

Introduction – lentiviral expression of open reading frames. Very vague.

References – not consistent in formatting, italicizing of species names, etc.

Point by Point response:

Reviewers' comments:

Reviewer #1 (Remarks to the Author):

The manuscript by Satchwell et al. presents use of a recently established immortalised erythroblast cell line, BEL-A, for infection by the human malaria parasite *Plasmodium falciparum*. The authors demonstrate that differentiated (enucleated) BEL-A-derived reticulocytes support *P. falciparum* invasion and intraerythrocytic development. The BEL-A line is further validated for use in the malaria community by demonstrating that genetic knock out of RBC receptors (Basigin) results in a block in parasite invasion. This phenotype can be rescued by complementation/re-introduction. The work described in this paper is promising as the BEL-A line provides a simplified and efficient system for RBC manipulation for the study of parasite phenotypes (and other purposes), compared to the use of erythroleukemic cells or in vitro maturation of hematopoietic stem cells. However, the manuscript does not go beyond validation and as such would be more suitable for a methods journal, unless more new biological insights are provided (including more detailed analysis of the discrepancy between the observed dispensability for CypB and previous work showing the contrary).

Major points

1. The authors provide only a superficial characterisation of *P. falciparum* growth and replication in BEL-A cells. There is no quantitative data on cycle length, PMR etc included. Instead a series of Giemsa smears are shown. To fully assess whether BEL-A cells constitute a suitable model to study host factors in *Plasmodium* development, more quantitative assessment of parasite propagation in these cells would be useful. In particular, addressing following questions would be interesting

- o Is the parasite multiplication rate (PMR) comparable between BEL-A cells and RBCs? The PMR could be easily calculated by determining the parasitemia before and after the second reinvasion.
- o How long can parasites be propagated in one batch of differentiated BEL-A cells? This could be addressed by culturing parasites over several cycles and assessing parasitemia/PMR in RBCs and BEL-A cells.

o Is the cycle length identical between RBCs and BEL-A cells, or are parasites delayed? From images in Figure 1 it appears that while in RBCs, some segmented schizonts are found at 46 hours, but in BEL-A cells, the schizonts are not yet segmented. Yet, this is obviously difficult to judge from a few selected Giemsa-stained parasites. A more detailed time course analysis could give further insight.

We have conducted additional experiments to provide the more detailed characterisation of invasion in the BEL-A derived reticulocytes requested by the reviewer. This includes measurement of the parasite multiplication rate and selectivity index between BEL-A derived reticulocytes, CD34⁺ cell derived reticulocytes and red blood cells (Figure 1 and Supplemental Figure 3). Notably, BEL-A derived reticulocytes exhibit no significant difference in their susceptibility to invasion and whilst the parasite multiplication rate is low in our assays (likely due to the low haematocrit used) no significant difference between the PMR of red blood cells, BEL-A or CD34⁺ cell derived reticulocytes is observed. Interestingly, both BEL-A derived reticulocytes and CD34⁺ cell derived reticulocytes exhibit a substantially higher selectivity index, indicating increased incidence of multiply invaded cells.

Regarding propagation of parasites over multiple cycles, we routinely see evidence of two rounds of invasion in our assays (illustrating the capacity for rupture and reinvasion). Since the assays performed were on a small scale (0.5×10^6 reticulocytes per assay) and did not involve supplementation of cultures with fresh BEL-A derived reticulocytes we were unable to track invasion cycles for more cycles alongside the large number of additional characterisation experiments performed, which would have been prohibitively costly. BEL-A derived reticulocytes kept refrigerated in PBSAG for up to four days were still susceptible to invasion (longer storage times were not tested). Extensive assessment of BEL-A derived reticulocyte stability in different storage solutions for longer periods has not been performed; however, given the demonstrated capacity for parasite reinvasion, we suggest that growth studies are likely to be dictated by the limits of cell numbers and economic considerations rather than inability to undergo multiple cycles of invasion.

A new detailed time-course illustrating equivalent progression of parasites through the respective stage of the intracellular lifecycle in BEL-A derived reticulocytes and red blood cells is now included in the new Figure 1.

2. Demonstration of CRISPR modification of BEL-A lines as a tool to investigate host factors of Plasmodium infection is exciting. However, validation of the method with the basigin knockout is a well-executed proof-of-concept. The authors continue to investigate a lesser-described putative receptor for Plasmodium, CypB and found it dispensable for invasion, in contrast to a previous study. Further in-depth analysis addressing the following points could enhance the biological insight of the paper.

- o The CypB knockout BEL-A line they investigate has a frame shift mutation, potentially leading to the expression of a truncated CypB that is not detected by Western blotting using the selected CypB antibody. While unlikely, there is the minor possibility that the expression of this truncated CypB is sufficient to support invasion of Plasmodium. Analysing a second CypB clone with a different mutation would be helpful to further clarify the role of CypB.

Since an additional clone based on the population transduced with the same guide would yield mutations at the same site within the protein coding sequence we do not believe analysing a second clone would be informative, particularly given the substantial amount of work and costs that this would involve. However, we have provided further validation of the absence of CypB in the knock out line described in the manuscript. The antigen against which the monoclonal antibody to cyclophilin B used in the original submission spans the region disrupted by the homozygous mutation. No putative truncation product was detected using this antibody however since the precise epitope is undefined this antibody may be unable to detect such a product. To address this, we have performed a western blot using a polyclonal antibody directed against an N terminal peptide entirely contained within the putative truncation product. This antibody detects full length cyclophilin B in unedited BEL-A cells but no wild type or putative truncation product in the CypB KO clone confirming the complete absence of protein. These new data are provided (Figure 4E).

- o The precise parameters under which the invasion experiments were performed are unclear. At which MOI was the infection done, i.e. how many schizonts were added to the 5×10^5 BEL-A cells to obtain the results depicted in Figure 4B? From the way it is written, it appears that two experiments were done, with lower and excess amounts of schizonts. Please show the data for both settings. If the data was obtained using excess schizonts, could it be that a putative mild phenotype of CypB deletion

is masked as the invasion capacity is saturated? It might be helpful to show dose-dependent effects using different MOIs.

Whilst for this experiment we do not have the precise MOI values, we have now presented data for high MOI (with correspondingly high invasion) and low MOI (Supplemental Figure 9). In both cases no significant difference in invasion of the CypB KO reticulocytes is observed compared to controls. Thus, we do not believe that a mild phenotype is being masked.

o In the study by Prakash et al., a peptide inhibitor of CypB successfully inhibited Plasmodium invasion. Have the authors tried these inhibitors on their WT and CypB KO BEL-A lines to test if the inhibitory effect of these peptides is specific/due to CypB?

We have not tried this experiment for several reasons. Firstly, as Prakash et al. state in their study, neither cyclosporin A (CsA) nor a novel peptide inhibitor is specific to cyclophilin B over other cyclophilins. CsA binds to multiple cyclophilins, and the novel inhibitor binds to cyclophilin A and perhaps other cyclophilins. In addition, CsA is known to have other, non-cyclophilin binding partners in red blood cells (see, for example, Ihara et al, 1990 *Transplant. Proc.* 22: 1736–1739). CsA may be quite non-specific (see, for example, Hu et al, 2004 *Bioinformatics*, Volume 30, Issue 24, Pages 3561–3566). Therefore, our CypB KO line contains multiple known binding partners for these drugs, which would convolute the results of this experiment.

Furthermore, in initial experiments in our laboratory, CsA was found to be poorly soluble, making such experiments practically challenging. Finally, BEL-A derived reticulocyte production and subsequent parasite invasion assays represent a significant time and cost investment, which, given the specificity and solubility concerns, we decided to devote to other experiments to improve the manuscript.

Additional comments

1. Results paragraph 5: “Expression of CypB in reticulocytes was undetectable by immunoblotting, however, a high level of expression was observed in undifferentiated BEL-A cells (Figure 3D).” According to figure legend and labeling, the blot in Figure 3D does not show any data from differentiated reticulocytes. Instead, it shows the expression of undifferentiated WT BEL-A cells compared to undifferentiated CypB-null BEL-A cells. If the labeling is correct, it should be referred to at the end of the following sentence: “Knockout of CypB was confirmed by immunoblotting of edited and unedited undifferentiated BEL-A cell lysates, confirming complete absence of detectable protein.” Please provide an immunoblot demonstrating the absence of CypB in differentiated reticulocytes.

We have replaced this figure with a new version that illustrates the expression of cyclophilin B in unedited BEL-A cells and absence in CypB KO in addition to the absence of detectable protein in reticulocytes derived from both unedited and edited BEL-A cells (new Figure 4E). Text and legend has been edited accordingly.

Could the authors also comment on why they hypothesize an involvement of CypB in Plasmodium invasion if CypB is not detectable in differentiated reticulocytes?

Prakash et al. previously reported expression of cyclophilin B at the surface of red blood cells and hypothesised the existence of a multiprotein complex between basigin and cyclophilin B, with

cyclophilin B identified as a crucial receptor for merozoite invasion of the red blood cell. The BEL-A cell line offers the unique opportunity to directly test this hypothesis within a cellular context by genetic deletion of the putative receptor. In the context of exploring the requirements of basigin for successful invasion, e.g. the prospective involvement of the cytoplasmic domain, determination of the effect of removal of this protein on the expression of basigin and on invasion efficiency was of interest. We could not detect CypB expression in BEL-A derived reticulocytes, hinting that CypB was not involved in invasion, and we confirmed this with invasion assays showing that no effect on invasion is observed upon deletion of this receptor. This illustrates how our cellular model can be used to interrogate hypotheses generated through less direct approaches.

2. Presentation of data in figure 2 is not in logical order, as initial validation of BEL-1 using the BSG KO is mixed with the subsequent analysis of CypB.

We understand the reviewer's point and agree that presentation of the inability of BSG KO reticulocytes to support invasion would be logically presented before subsequent rescue experiments and assessment of the requirement for CypB. However, for practical reasons the experiments (KO and complementation as well as CypB KO) were performed in parallel within the same experiment allowing for relative invasion rates to be measured compared to a control as well as each other. For this reason and for clarity we are most comfortable with presentation of the invasion statistics together (as they were performed experimentally) amalgamated in the same figure rather than separated out since they are normalised to the same control.

3. Figure 3B: The basigin western blot does not show a clear band, but instead a smear across different molecular weights. While the absence of basigin in the knockout reticulocytes is obvious, it is difficult to see a shift towards a lower molecular weight for the truncated BSG Δ C. It would be helpful to repeat this western blot to achieve a clearer result and higher resolution, and also to provide the expected migration pattern of full-length and truncated basigin.

Basigin is a highly glycosylated protein that runs as a smear on an SDS PAGE gel. The C terminal truncation is relatively short resulting in a mild shift in the leading edge of this smear that whilst subtle is discernible. Nevertheless, to provide additional evidence that this signal corresponds to the truncated basigin product we have provided an additional figure in which lysates of reticulocytes derived from unedited BEL-A cells, WT BSG rescue and BSG Δ C rescue have been loaded. Immunoblotting with an antibody with epitope specific to the C terminus of basigin detects endogenous and wild type rescue basigin but no signal is observed in the BSG Δ C expressing cells (Figure 4C).

4. Figure 4C and results, paragraph 8: The depicted flow cytometry plots lack a quantification. In particular the statement in paragraph 8 of the results section "Quantification of invasion efficiency into modified reticulocytes replicated results obtained through manual counting" requires quantitative data of triplicate experiments. Is there a reason that the rightmost population of SybrGreen-positive cells is excluded from the gate? As a control for the background BEL-A cells give in flow cytometry, wouldn't it be possible to subtract the signal of the negative control (Heparin-treated BEL-A cells) from the other experimental conditions? It would also be informative to see how uninfected RBCs and BEL-A cells look like in this flow analysis.

We have restructured the manuscript to present a robust validation of flow cytometric quantification of invasion efficiency in unedited BEL-A derived reticulocytes as a new Figure 2. Requested controls and uninfected cell plots are included. Figure 4C flow cytometry data has been removed - data for quantification of invasion in Figure 4B (new Figure 5B) is derived from manual counts of cytopins from triplicate experiments.

- Introduction, paragraph 2: Bracket missing

Manuscript altered as requested

- Introduction, paragraph 4: “the significant membrane complex remodeling and loss of protein (basigin and CD44 in particular)”. This phrase sounds misleading, suggesting that no basigin is left on the reticulocyte after enucleation, which would contradict the essentiality of basigin for Plasmodium invasion. Clarify that not all basigin is lost during enucleation.

Manuscript altered as requested

- Results paragraph 3: “To assess the capacity to successfully rescue the invasion defect brought about by the absence of basigin...”. This invasion defect has not been shown yet in the paper. The results demonstrating an invasion defect in the BSG KO are presented only in paragraph 6.

Manuscript altered as requested

- Results paragraph 3: The reasoning for complementing the knockout with a truncated BSG does not become clear. The hypothesis that the cytoplasmic tail is involved in signaling, thus mediating Plasmodium invasion, is only mentioned in the discussion section. Please add a short note here as to why BSG Δ C was investigated.

Manuscript altered as requested

- Results, paragraph 6: Please refer here already to Figure 4B as it depicts the results of the manual counting of invasion.

Manuscript altered as requested

- Materials and Methods, Flow Cytometry: In the buffer description of PBSAG, BSA is mentioned twice.

This description refers to the fact that the PBSAG wash buffer is supplemented with an additional 1% BSA for flow cytometry labelling.

- Figure legend Figure 2: “Flow cytometry histogram illustrates absence of basigin (HIM6) labelling in reticulocytes derived from unedited (blue) and basigin knockout (red) BEL-A cell lines compared to IgG isotype control (grey).” Incorrect description, as basigin labeling is present, not absent, in unedited BEL-A cell lines (as expected). Please correct.

Legend altered as requested

Reviewer #2 (Remarks to the Author):

In their manuscript “Stable knockout and complementation of receptor expression using in vitro cell line derived reticulocytes for dissection of host malaria invasion requirements”, Satchwell and colleagues reports the feasibility of using an immortalized erythroblast cell line for differentiating into reticulocytes compatible for invasion of *Plasmodium falciparum*. They also demonstrate potential applications of this system through CRISPR-mediated knock-out of a well known host receptor, basigin leading to invasion inhibition. Authors state several advantages of BEL-A derived reticulocytes for plasmodium research over other existing platforms (such as erythroblasts derived from JK-1 cells), which all appear convincing to this reviewer making it suitable for publication in Nature Communications. This is impactful work, with potential implications in the study of *P. vivax* biology and invasion mechanisms.

We thank the reviewer for their positive appraisal of the manuscript.

Remarks

1. For invasion efficiency comparison (Figure 1), authors matched BEL-A derived reticulocytes to native red blood cells and reports a ratio of 1.08:1 for the number of invasion events. I have to assume that native red blood cells refers to peripheral blood collected from a hospital or blood bank (this information could not be found in the methods section). In such a case, comparison is made between BEL-A derived reticulocytes to mature normocytes which brings some discrepancy. As such, reticulocytes are invaded by most species of plasmodium (including *P. falciparum*) with higher affinity compared to normocytes, with at least a 2-fold higher efficiency for the former. I would therefore recommend the authors to include a comparison between BEL-A derived reticulocytes, normal healthy reticulocytes (purify them from cord blood or peripheral blood using CD47-conjugated magnetic beads or density gradients) and mature normocytes either as part of Figure 1 or as supplemental data.

The red blood cells used in the manuscript are indeed peripheral blood RBCs collected from a blood bank. To address the reviewer’s request, we attempted to purify reticulocytes from peripheral blood using magnetic CD71 beads for comparison to red blood cells and BEL-A derived reticulocytes; however, we found that the bead-based isolation procedure, whilst allowing for very high levels of purity, caused reticulocytes to clump together. Invasion efficiency in these bead-isolated reticulocytes was thus significantly lower than that in red blood cells or BEL-A derived reticulocytes, an effect we attribute to this clumping. Therefore, to compare invasion efficiency of BEL-A derived reticulocytes to another source of reticulocytes as well as red blood cells, we generated reticulocytes derived through *in vitro* culture of primary CD34⁺ cells (which have been extensively used within the literature for invasion studies e.g. Crosnier et al, Egan et al). Figure 1 now includes data comparing parasite multiplication rate and selectivity index for each cell type. Whilst we did not observe increased invasive susceptibility of reticulocytes derived from BEL-A cells or CD34⁺ cells compared to red blood cells in our assays, BEL-A derived reticulocytes were invaded as efficiently as those derived from primary CD34⁺ cells.

2. Figure 1: ability of BEL-A derived reticulocytes to support growth and re-invasion of *falciparum*: 38h, 46h and 62h post invasion. Can the authors elaborate on the invasion efficiency in the second cycle compared to the first (62h post invasion)?. Was the invasion rate higher or lower than the first cycle? During the course of this experiment, does all the non-infected reticulocytes stay as immature reticulocytes or a small fraction mature into more mature reticulocytes (CD71 negative) or even

normocytes? This is subjective, but an immunofluorescence-based assay of CD71 expression as surrogate marker for immature reticulocytes could be adopted.

We have some concerns about rigorous quantification of the second cycle invasion, primarily because the different cell types might have different levels of lysis over 62 hours, which would influence the measured parasitemia. We would therefore recommend that the first cycle is used for quantification of invasion by manual counting or flow cytometry. Nevertheless, we did calculate the parasite multiplication ratio (PMR) in the second cycle using manual counting (n=2). The data are shown below, alongside the PMR for the first cycle. There were no significant differences seen in the PMRs across the three cell types, red blood cells, and BEL-A and CD34+ derived reticulocytes. Furthermore, the PMR is similar across the two cycles for the BEL-A derived reticulocytes (with a value of 0.6).

We did not monitor the expression of CD71 in our invasion assays to track maturation of reticulocytes. Whilst this is an interesting question we feel that it is tangential to this study, and given the large amount of characterisation experiments that needed to be performed with limited material, would be best explored through future work.

3. Were there more multiple infections in reticulocytes compared to normocyte controls?

We have performed additional experimental work to quantify this and interestingly, observe significantly more multiple infections in reticulocytes derived from either BEL-A cells or CD34+ cells compared to mature red blood cells. This data is presented in the new Figure 1 and Supplemental Figure 3.

4. Page 4: Excess purified schizonts were added to 5×10^5 target cells resulted in ~60% parasitemia in unedited cells, compare to no rings in BSG KO reticulocytes. What is the excess amount of schizonts to target cells?

While the exact initial schizont-specific parasitemia was not quantified in this particular high MOI experiment, we have performed new experiments in which we quantified both the initial "schizontemia" and resultant parasitemia post-invasion of unedited BEL-A derived reticulocytes, allowing calculations of the parasite multiplication rate (PMR) (Figure 1, Supplemental Figure 3). Three independent experiments were set up at high (approximately 17% schizonts added) and low MOI (approximately 5% schizonts added), resulting in the ring stage parasitemias shown in the new Supplemental Figure 2. The PMRs for invasion into red blood cells, BEL-A derived cells, and CD34+ derived cells, are all calculated to be approximately 0.6 (Figure 1 and Supplemental Figure 3). This value is lower than published values (see e.g. Kanjee et al.), which we believe is because of the low

haematocrit at which these assays were performed. Low haematocrit was necessary due to limiting cell numbers.

5. The authors claims that the flow cytometry method used in this study was capable of rapid initial assessment of invasion in BEL-A derived reticulocytes. However it is best applied only when parasitemia is high. The flow plot clearly shows residual background in BSG KO cells some of which may be due to post-leukofiltration contamination, and un-ruptured schizonts. However, extreme care needs to be taken in drawing conclusions from such experiments as high background from non-infected cells that stains partially positive for SYBR-green. I would still recommend manual counting of Giemsa-stained smears to confirm the invasion efficiency.

We agree with the reviewer's comments and tried in our initial submission to highlight the care that should be taken in interpretation of flow cytometric invasion assays of this type. We now present in Figure 2 a robust appraisal of flow cytometric quantification of invasion efficiency using SYBR Green in BEL-A derived reticulocytes. Nevertheless, as highlighted in the manuscript, manual counting of Giemsa stained smears remains the gold standard, allowing for exclusion of contaminants; therefore, since manual counting was the means by which we obtained the triplicate invasion efficiency data in Figure 4B (now 5B), we have removed the supportive flow cytometry data in Figure 4C, providing clarity on the source of the invasion statistics.

6. Flow cytometry plots as the final component of Fig. 4 in some ways deviates the attention of a reader from main conclusions of the paper which is viability of BEL-A reticulocytes for functional studies in plasmodium. Authors may want to consider re-structuring the sequence of these figures.

We agree with the reviewer and thank them for this suggestion. We have restructured the manuscript, emphasising the main conclusions as recognised in this comment.

Reviewer #3 (Remarks to the Author):

The manuscript by Satchwell et al demonstrates that in vitro differentiation of an enucleation competent immortalized erythroblast cell line (BEL-A) supports the growth of Plasmodium falciparum and that CRISPR/Cas9 editing technology as well as lentiviral expression of transgenes can be used to dissect the role of host receptors in P. falciparum invasion. As such, this provides a very important advance for researchers working on the blood stages of Plasmodium because it now provides a mechanism to determine the role of host proteins in supporting growth and survival of the parasite, which have traditionally relied on much less accessible or more time-consuming approaches (eg. Knockout mice, using donor RBCs from patients that are deficient or have defective host protein of interest, etc). In this study the authors used this system to validate previous findings using several other approaches that basigin is an essential host receptor for Plasmodium invasion and second, that contrary to another study, cyclophilin B is not essential for invasion.

We thank the reviewer for their positive comments in support of the manuscript.

Nevertheless, the study could do with some greater rigor in parts to support their conclusions. Other comments I have provided to improve the flow or clarity of the manuscript.

1. Abstract – The abstract is a bit clunky in parts and could do with rewording to make it clearer. Eg. Using CRISPR-mediated gene knockout and lentiviral expression of open reading frames. This is

vague. It would be clearer to say using CRISPR-mediated gene knockout and subsequent complementation of the open reading frame of the gene of interest via lentiviral expression. I would also split up the sentence 'rescued by re-expression of the receptor' and 'mutant thereof' and introduce the receptor mutant in the following sentence. Eg. Specifically, the ability *P. falciparum* to invade the edited clone complemented with a basigin mutant lacking its cytoplasmic domain excludes a role for this domain during basigin invasion.

We have provided a new (<150 words) abstract reducing the complexity of the abstract and providing clarity as requested.

2. The authors have demonstrated that the BEL-A line can support *P. falciparum* growth. Because this is the first time this is reported it would be useful to provide further insight. How many generations can the parasite grow in the culture, have they looked beyond the cycle after reinvasion (ie. Beyond 62hr)? From Fig 1 it looks like the parasites growing in BEL-A derived reticulocytes have developed more quickly and are more mature at 14 hr and even 38 hr, yet schizonts have not fully-developed by 46 hr when compared to mature RBC. Is the timing of parasite development/stages different between mature RBC and BEL-A derived reticulocytes? Could this be because they are growing in retics rather than more mature RBC or because of the cell line. How many schizonts were added to the cells (ie. What was the ratio?) and what levels of invasion (% parasitemia) were observed?

We have provided a new more detailed time-course of parasite development in BEL-A derived reticulocytes compared to red blood cells, which illustrates equivalent development of *P. falciparum* through the definitive stages of the life cycle in both cell types (Figure 1A-B). As described above, we were unable for practical reasons to analyse invasion over further cycles.

We have performed new experiments to calculate the parasite multiplication rate (PMR), or the ratio of resultant invaded ring stage parasites to the schizonts added in the assay, as shown in new Figure 1D and Supplemental Figure 3. As described above in response to reviewer 1, we set up three independent experiments at high MOI (approximately 17% schizonts) and low MOI (approximately 5% schizonts). The ring stage parasitemias resulting from each MOI are shown in new Supplemental Figure 2.

3. Structure of the article- for the sake of flow, it would have been best after BEL-A BSG null clones were generated to assess at this point the ability of parasites to invade and hence validate that basigin is essential for invasion before doing complementation studies to look at the role of particular domains (ie. At bottom of page 3 of manuscript).

As detailed in response to reviewer 1, we understand the reviewer's point and agree that presentation of the inability of BSG KO reticulocytes to support invasion would be logically presented before subsequent rescue experiments and assessment of the requirement for CypB. However, for practical reasons, the experiments (KO and complementation as well as CypB KO) were performed in parallel within the same experiment, allowing for relative invasion rates to be measured compared to a control as well as each other. For this reason and for clarity we are most comfortable with presentation of the invasion statistics together (as they were performed experimentally) amalgamated in the same figure rather than separated out since they are normalised to the same control.

4. 1st paragraph page 4 – The last two sentences are a bit confusing. I would reword to the effect that you cloned full-length and truncated BSG sequences into the lentiviral pLVX-Neo for expression in BEL-A cells such that the guide sites were mutated so that they were resistant to editing.

Manuscript adjusted as requested

5. Role of basigin and C-term domain (Figure 3). What is the variation in basigin expression in the control (Fig 3A)? The western blot with basigin antibody (Figure 3B) is difficult to decipher with so many bands and to me there doesn't appear to be any visible difference between wildtype and the truncation mutant to demonstrate it is indeed truncated. The truncation mutant needs to be confirmed in another manner in order to draw the conclusion that the C-term is not essential for invasion. What about transcript size to demonstrate expected transcript length? In Fig 3C. Is difference in GPC between control and CypB KO statistically significant?

Data presented in Figure 3A (now Figure 4A) for expression of basigin in the various edited BEL-A line derived reticulocytes is normalised to the expression of endogenous basigin in reticulocytes derived from unedited BEL-A cells for that particular experiment, with error bars indicating variation across independent experiments. The data was normalized in order to correct for variation between repeats, given that the experiments were separated over the course of many months which also included changes in machine settings. Therefore, no error bars can be added to the control, since the process of normalization inherently sets the expression in unedited cells to 100%.

Regarding the western blot in Figure 3B (now Figure 4B), as highlighted in our response to reviewer 1, basigin is a highly glycosylated protein that runs as a smear on an SDS PAGE gel. The C-terminal truncation is relatively short, resulting in a mild shift in the leading edge of this smear that whilst subtle is discernible. Nevertheless, to provide additional evidence that this signal corresponds to the truncated basigin product, we have provided an additional figure in which reticulocytes derived from unedited BEL-A cells, WT BSG rescue and BSG Δ C rescue have been loaded. Immunoblotting with an antibody with epitope specific to the C terminus of basigin detects endogenous and wild type rescue basigin but no signal is observed in the BSG Δ C expressing cells (Figure 4C).

In Figure 3C, the difference between expression of GPC in unedited and CypB KO is a small (<20%) but statistically significant reduction; however, given that the wild-type BSG rescue also exhibits an equivalent statistically significant reduction in GPC expression, we do not believe this to be important.

6. Contribution of Cyclophilin B to invasion. That expression of CypB in reticulocytes is undetectable is important when weighing up if this protein is an important receptor for Plasmodium invasion. Presumably it is also absent from mature RBC. It would be good to have a blot or qRT-PCR to look at expression at the different stages. Importantly, the full blot for D should be included to rule out whether a truncated product is still expressed and this should be validated using another approach if the antibodies bind downstream of where the truncation takes place. Why were the FACS plots not performed as per Fig 2 with cyclophilin B antibody in place of basigin antibody?

We are unable to detect CypB by immunoblotting or flow cytometry in reticulocytes or red blood cells, respectively, and whilst we detect robust expression in undifferentiated BEL-A cells (proerythroblast morphology) no surface expression is detectable by flow cytometry. To address the reviewer's request for expression data during erythropoiesis, we have added a Supplemental Figure (Supplemental Figure

7) generated using the proteomic dataset and database published by Gautier and colleagues (Cell Reports, 2016), which illustrates a progressive and dramatic reduction in copy number of CypB during erythroid differentiation.

We have provided the full blot for Figure 3D (now Figure 4E) as requested. Note that the monoclonal antibody used in the original submission was raised against an antigen that spans the region disrupted by the homozygous mutation. No putative truncation product was detected using this antibody; however, since the precise epitope is undefined, this antibody may be unable to detect such a product. To address this, we have performed a western blot using a polyclonal antibody directed against an N terminal peptide entirely contained within the putative truncation product. This antibody detects full-length cyclophilin B in unedited BEL-A cells, but no wild type or putative truncation product in the CypB KO clone confirming the complete absence of protein.

Flow cytometry using the cyclophilin B antibody was performed on red blood cells and undifferentiated BEL-As but no positive signal was detected. These plots are now included as Supplemental Figure 6.

7. Assessment of invasive susceptibility of modified reticulocytes – how many schizonts were added, what was the ratio of schizont to reticulocytes, and how many cells were manually counted. Fig 4 A should include the unedited control. It would be helpful if Fig 4B showed the SD of % invasion of the control (ie. expressed as 100% but still shows SD).

As detailed above, we have performed new experiments to quantify the parasite multiplication rate (PMR), now shown in Fig 1D and Supplemental Figure 3. Fig 4A has now been updated to include unedited BEL-A controls (new Figure 5A). We are grateful to the reviewer for flagging this, as in remaking the figure, we discovered that the labels were swapped on the BSG KO + BSG WT and BSG KO + BSGΔC cell lines. This has now been corrected in the new Figure 5A.

1,000 cells were manually counted. For each replicate, the invasion was normalised to the percentage invasion of the control, giving 100% invasion in each case; therefore, there is no standard deviation of the control.

8. The FACS analysis is important to perform because it is quantitative and can be used to assess a large number of cells to determine whether even very low levels of invasion have occurred in the BEL-A edited lines +/- complementation. Unfortunately the FACS has not been robustly performed. I disagree with the authors that FACS is reliant on the absence of DNA in uninfected RBC and nucleated erythroid cells can be used if a different approach is used. For example, one can discriminate between nucleated uninfected cells and infected reticulocytes or nucleated cells if GFP-expressing parasites are used for invasion, in addition to stains. Presumably the nucleated cells are also much bigger and FSC/SSC could be used to discriminate these as could staining for human equivalent to Ter119/CD44-CD71 as these would differ in maturation stages of erythroid cells. Whilst SYBR green stains reticulocyte RNA, Hoechst could also be used (the nucleated cell most likely gives much higher levels of fluorescence than infected reticulocyte). As it is the authors do see events in the invaded gates but they have to rely on cytopins to say these are nucleated cells and therefore discount them. Also please include the % invaded on the plots in Fig 4C. No mention is made of what the events are that are to the right of the invaded gate? Moreover, what are the 2 peaks in the invade gate?

We thank the reviewer for their insights and suggestions regarding flow cytometric quantitation of invasion. We have in fact previously attempted to quantify invasion using GFP expressing parasites without nucleic acid staining and found invaded cells difficult to reliably discriminate above the autofluorescence of the uninfected cells. Similarly, on a FSC/SSC plot, late orthochromatic erythroblasts and to some extent pyrenocytes (extruded nuclei) are not sufficiently different in size to separate from reticulocytes based on these parameters alone. Differences in expression of markers such as GPA, CD44 and CD71 between residual orthochromatic erythroblasts and reticulocytes in unedited cells is not large enough to reliably use for gating purposes and cannot be applied to experiments in which expression of such markers is altered; moreover, this approach complicates the assay significantly.

We agree with the reviewer that Hoechst (appropriately excited with a UV laser) would be preferable to SYBR Green due to its higher specificity for DNA; however, Hoechst is inefficiently excited by the 405nm laser available to us. We do not have access to a flow cytometer with a UV laser, and such machines are not widely accessible, hence our use of SYBR Green for these studies. The phrase in question referred specifically to the use of nucleic acid staining dyes for flow cytometry (confounded in nucleated cell lines) rather than flow cytometry itself, we have amended this statement for clarity.

We have also restructured the manuscript to provide a more robust illustration of the capacity for quantification of invasion in BEL-A derived reticulocytes using flow cytometry, with % invasion labelled for singly and multiply invaded populations highlighted in the new Figure 2. We show that invasion parameters such as the parasite multiplication rate and selectivity index can be calculated by flow cytometry to generate comparable values to manual counting (Supplemental Figure 3).

Discussion.

RhopH3 has been shown in two studies using conditional gene knockout that it is essential for parasite invasion of RBC. Hence the authors should change the last sentence of paragraph 2 page 6 as their work has just shown that cyclophilin B is not essential. It may be that this interaction between RhopH3 and cyclophilin is not a true interaction.

We agree with the reviewer and have altered the sentence accordingly.

The authors mention that this system could be used to look at parasite development and egress, however, it would be worthwhile pointing out the obvious caveat that knocking out these proteins in the undifferentiated BEL-A line may have dire consequences and may not support differentiation and enucleation.

It is true that generation of novel reticulocyte phenotypes is always reliant upon the capacity for edited cell lines to undergo terminal differentiation and enucleation. As such, thorough characterisation of edited lines and validation of capacity for enucleation is an important precursor to any invasion studies. We have added a comment to the manuscript to highlight this.

Other comments:

Introduction – lentiviral expression of open reading frames. Very vague.

Manuscript adjusted as requested

References – not consistent in formatting, italicizing of species names, etc.

Manuscript adjusted as requested

Reviewers' Comments:

Reviewer #1:

Remarks to the Author:

In this revised manuscript the authors have addressed all comments raised by this reviewer. A couple of suggestions for discussion in the text are listed below.

- Inclusion of CD34+ reticulocytes for PMR and invasion assays is very informative. Notably, both CD34+ retics and BEL-A show increased selectivity compared to normocytes. This is similar to the findings by Kanjee et al with the JK-1 line. Could the authors speculate as to why that is? This may be a suitable point for the discussion.

- In the same experiment PMR is measured and shown to be below 1 for all 3 lines, probably due to the low hematocrit used. Nevertheless there is still no formal demonstration that parasites can grow continuously in this system, which would represent a key advantage over published lines such as JK-1. The authors mention the high costs of maintaining the BEL-A cells as a reason why this was not tested rigorously. These possible limitations (lack of efficient continuous growth, cost considerations) should be mentioned in the discussion.

Matthias Marti

Reviewer #2:

Remarks to the Author:

Authors did a good job and addressed all comments in the revisions. I am happy to recommend publication of this paper in its current (R1) form.

Reviewer #3:

Remarks to the Author:

The authors have taken my comments on board and the changes they have made to the manuscript are satisfactory.

I just have a question with respect to Figure 4D versus Supp Fig 5 and the comment on Line 246, where the authors state there is a mild reduction in GPC in the CypB knockout. The FACS plot in Supp Fig 5 suggest that of GPC is similar but that of CD44 is reduced. I was wondering if that of CD44 is significant?

Given that several of the reviewers questioned about what happens to invasion over subsequent cycles, it would be worthwhile making a comment in manuscript about why this was not undertaken as readers of the manuscript will no doubt be asking the same question.

Response to Reviewers

Reviewer #1 (Remarks to the Author):

In this revised manuscript the authors have addressed all comments raised by this reviewer. A couple of suggestions for discussion in the text are listed below.

We thank the reviewer for their positive appraisal of the manuscript

- Inclusion of CD34+ reticulocytes for PMR and invasion assays is very informative. Notably, both CD34+ retics and BEL-A show increased selectivity compared to normocytes. This is similar to the findings by Kanjee et al with the JK-1 line. Could the authors speculate as to why that is? This may be a suitable point for the discussion.

We agree that this is an interesting observation and have addressed this in the discussion as requested

- In the same experiment PMR is measured and shown to be below 1 for all 3 lines, probably due to the low hematocrit used. Nevertheless there is still no formal demonstration that parasites can grow continuously in this system, which would represent a key advantage over published lines such as JK-1. The authors mention the high costs of maintaining the BEL-A cells as a reason why this was not tested rigorously. These possible limitations (lack of efficient continuous growth, cost considerations) should be mentioned in the discussion.

A comment has been added to the discussion

Matthias Marti

Reviewer #2 (Remarks to the Author):

Authors did a good job and addressed all comments in the revisions. I am happy to recommend publication of this paper in its current (R1) form.

We thank the reviewer for their positive appraisal of the manuscript

Reviewer #3 (Remarks to the Author):

The authors have taken my comments on board and the changes they have made to the manuscript are satisfactory.

I just have a question with respect to Figure 4D versus Supp Fig 5 and the comment on Line 246, where the authors state there is a mild reduction in GPC in the CypB knockout. The FACS plot in Supp Fig 5 suggest that of GPC is similar but that of CD44 is reduced. I was wondering if that of CD44 is significant?

A variably reduced expression of CD44 was observed in the CypB KO. Due to the relatively low expression of CD44 compared to GPC and the log scale in Supplementary Figure 5 this appears more evident. However, across the three independent experiments this reduction compared to unedited cells was not statistically significant.

Given that several of the reviewers questioned about what happens to invasion over subsequent cycles, it would be worthwhile making a comment in manuscript about why this was not undertaken as readers of the manuscript will no doubt be asking the same question.

We agree that this is an important point and have added a comment to the discussion in the main manuscript.